# Thermodynamic constraints on the assembly and diversity of microbial ecosystems are different near to and far from equilibrium

**Jacob Cook**[1,2]*, **Samraat Pawar**[3☯], **Robert G. Endres**[1,2☯]*

**1** Department of Life Sciences, Imperial College London, London, United Kingdom, **2** Centre for Integrative Systems Biology and Bioinformatics, Imperial College London, London, United Kingdom, **3** Department of Life Sciences, Imperial College London, Silwood Park Campus, Ascot, United Kingdom

☯ These authors contributed equally to this work.
* j.cook17@imperial.ac.uk (JC); r.endres@imperial.ac.uk (RGE)

**Data Availability Statement:** The code used to generate the results shown in this paper can be accessed from the following GitHub repository

## Abstract

Non-equilibrium thermodynamics has long been an area of substantial interest to ecologists because most fundamental biological processes, such as protein synthesis and respiration, are inherently energy-consuming. However, most of this interest has focused on developing coarse ecosystem-level maximisation principles, providing little insight into underlying mechanisms that lead to such emergent constraints. Microbial communities are a natural system to decipher this mechanistic basis because their interactions in the form of substrate consumption, metabolite production, and cross-feeding can be described explicitly in thermodynamic terms. Previous work has considered how thermodynamic constraints impact competition between pairs of species, but restrained from analysing how this manifests in complex dynamical systems. To address this gap, we develop a thermodynamic microbial community model with fully reversible reaction kinetics, which allows direct consideration of free-energy dissipation. This also allows species to interact via products rather than just substrates, increasing the dynamical complexity, and allowing a more nuanced classification of interaction types to emerge. Using this model, we find that community diversity increases with substrate lability, because greater free-energy availability allows for faster generation of niches. Thus, more niches are generated in the time frame of community establishment, leading to higher final species diversity. We also find that allowing species to make use of near-to-equilibrium reactions increases diversity in a low free-energy regime. In such a regime, two new thermodynamic interaction types that we identify here reach comparable strengths to the conventional (competition and facilitation) types, emphasising the key role that thermodynamics plays in community dynamics. Our results suggest that accounting for realistic thermodynamic constraints is vital for understanding the dynamics of real-world microbial communities.

https://github.com/jacobcook1995/
EcosystemAssembly/tree/master/Assembly.

**Funding:** JC was supported by a PhD studentship awarded by the Natural Environment Research Council (NERC) CDT in Quantitative and Modelling Skills in Ecology and Evolution (grant No. NE/P012345/1). In addition, SP was supported by Leverhulme Research Fellowship (RF-2020-653\2) and NERC grant NE/S000348/1. The funders had no role in study design, data collection and analysis, decision to publish, or preparation of the manuscript.

**Competing interests:** The authors have declared that no competing interests exist.

## Author summary

There is a growing interest in microbial communities due to their important role in bio-geochemical cycling as well as plant and animal health. Although our understanding of thermodynamic constraints on individual cells is rapidly improving, the impact of these constraints on complex microbial communities remains largely unexplored theoretically and empirically. Here, we develop a new microbial community model which allows thermodynamic efficiency and entropy production to be calculated directly. We find that availability of substrates with greater free-energy allows for a faster rate of niche generation, leading to higher final species diversity. We also show that when the free-energy availability is low, species with reactions close to thermodynamic equilibrium are favoured, leading to more diverse and efficient communities. In addition to the conventional interaction types (competition and facilitation), our model reveals the existence of two novel interaction types mediated by products rather than substrates. Though the conventional interactions are generally the strongest, the novel interaction types are significant when free-energy availability is low. Our results suggest that non-equilibrium thermodynamics need to be considered when studying microbial community dynamics.

## Introduction

The constraints thermodynamics place upon on individual organisms inevitably impact ecosystem dynamics. Thus, attempts to understand ecosystems through thermodynamic principles have been made repeatedly throughout the history of ecology [1–3]. These attempts have generally involved the development of coarse, whole-ecosystem level extremal (maximisation or minimisation) principles, such as flux [1] and power maximisation [2]. Most notable of these is the maximum entropy production principle, that ecosystems tend towards states that produce entropy at the maximum achievable rate [3]. This principle has been applied to some degree of success to predicting ecosystem characteristics such as spatial distribution of vegetation [4] and biogeochemical cycling in ponds [5]. Though consideration of ecosystem wide entropy production has led to insights in areas such as the conditions for ecosystem stability [6], without detailed consideration of the underlying mechanisms, its explanatory potential remains limited. However, when these mechanisms have indeed been considered, they are implicitly assumed to be dissipating sufficient free-energy that thermodynamic constraints can reasonably be neglected. In contrast, in biophysics, non-equilibrium thermodynamics at the cellular level have been considered in much greater detail [7], particularly in the areas of kinetic proofreading [8] and sensing accuracy [9, 10]. This opens the possibility of detailed consideration of the impact of thermodynamic constraints on ecosystem dynamics.

Thermodynamic constraints are especially pertinent to microbial community dynamics because microbial growth can be described explicitly in terms of the energetics of carbon substrate processing. Microbes experience a large number of physical constraints on their metabolism (see [11] for a review). Here, we consider two universally-relevant constraints. First, the widely studied constraint of a finite cellular proteome which means that the increased expression of a particular class of metabolic protein must occur at the expense of other metabolic proteins and/or ribosomes [12–14]. Second, in order to proceed rapidly metabolic reactions must (net) dissipate substantial amounts of free-energy. This both limits the set of possible (catabolic) reactions that microbial cellular metabolic networks can be formed from, and leads to a significant reduction in reaction rates close to thermodynamic equilibrium. Both these constraints affect microbial growth rates and ultimately microbial interactions, but the impact

of the thermodynamic constraint has rarely been considered previously. Models of the impact of thermodynamic inhibition have been developed to properly capture the growth rate of anaerobic microbial populations [15]. This has been recently extended to study whether thermodynamics leads to distinct ecological strategies [16], to explain the coexistence of multiple species on a single substrate [17], and to explain empirically observed limitations on the growth of simple methanogenic communities [18]. However, these studies have considered relatively simple systems of a few interacting species.

The carbon substrates that microbes feed upon are often classified in terms of the difficulty involved in breaking them down, with those easily broken down being termed 'labile', and ones that are hard to break down termed 'recalcitrant'. In energetic terms, a recalcitrant substrate can be thought of as one that requires a significant energy investment by a microbe for a minimal return. This could emerge through multiple mechanisms such as high substrate activation energies, substrates that have to be broken down extracellularly, or reactions that yield small amounts of free-energy. Thus, in a thermodynamic model, recalcitrance can be approximately modelled by considering substrates of low free-energy, potentially shedding light on the role of substrate lability versus recalcitrance on the dynamics of microbial community assembly.

Here, we develop a mathematical model that explicitly and mechanistically incorporates thermodynamic constraints. We use this model to study the emergent impact of thermodynamic constraints on microbial community diversity by simulating their assembly. Because natural communities may assemble on a diversity of substrates, we focus on the relationship between microbial strain (species) efficiency, substrate free-energy, and emergent species diversity. We find free-energy availability to be the single most important driver of community diversity, as greater energy availability allows for a faster rate of niche generation. We also show that consideration of thermodynamic constraints leads to new insights into the fundamental mechanistic and energetic basis of species interactions.

## The model

Our thermodynamic microbial community model is a thermodynamically-explicit extension of the MacArthur consumer-resource model [19]. Its key features are illustrated in Fig 1 with a detailed derivation given in S1 Appendix. In this model, catabolic reactions are explicitly modelled, but anabolism is abstracted as protein translation. As the predominant energy cost for microbial cells is protein translation [20], the implicit assumption that this is the only energetic cost is a reasonable one. Each metabolite ($\beta$) is represented by its concentration $C_\beta$ and each consumer ($i$) is represented by three variables: its population abundance $N_i$, ribosome fraction $\phi_{R,i}$, and internal energy (ATP) concentration $a_i$. The dynamics of these variables are given by

$$\frac{dC_\beta}{dt} = \kappa\delta_{\beta,1} - \rho C_\beta + \sum_{i=1}^{B}(p_{i,\beta}(\mathbf{C}) - c_{i,\beta}(\mathbf{C}))N_i \tag{1}$$

$$\frac{dN_i}{dt} = (\lambda_i(a_i, \phi_{R,i}) - d_i)N_i \tag{2}$$

$$\frac{d\phi_{R,i}}{dt} = \frac{1}{\tau_g}\left(\phi_R^*(a_i) - \phi_{R,i}\right) \tag{3}$$

$$\frac{da_i}{dt} = J_i - \chi m\lambda_i - a_i\lambda_i, \tag{4}$$

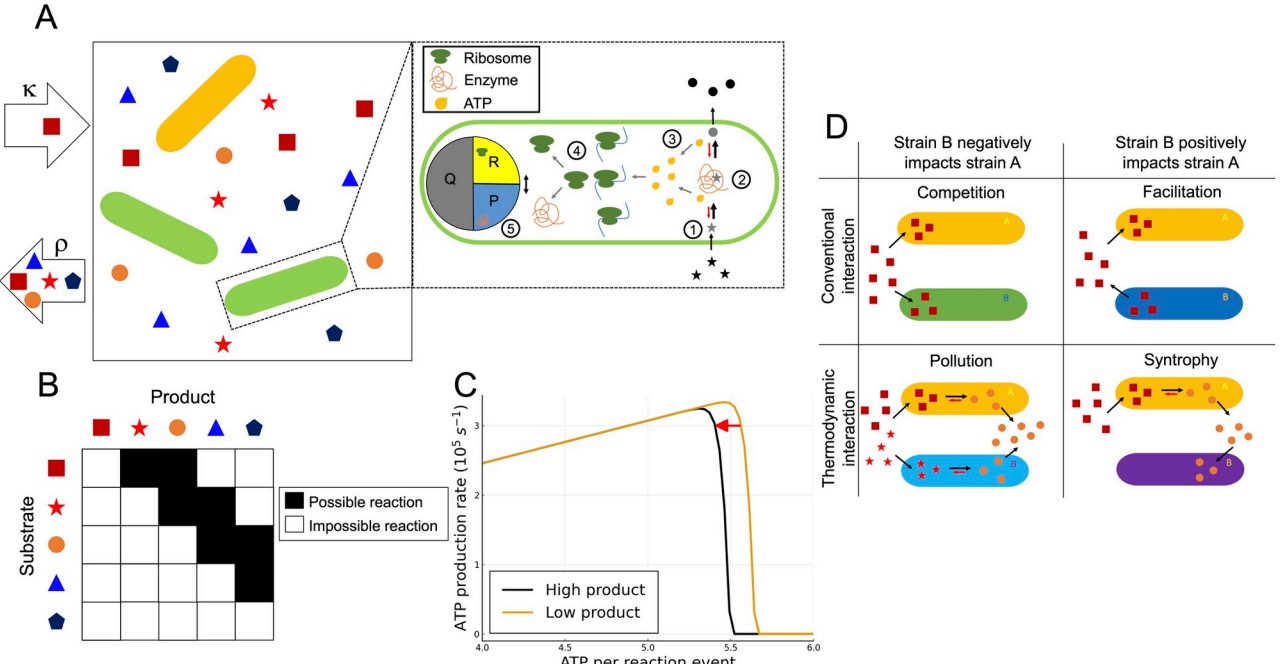

**Fig 1. Overview of the thermodynamic microbial community model. A**: A single substrate is supplied with rate $\kappa$, and diversifies into secondary substrates (metabolites) through uptake, metabolism and leakage by the microbial populations. All metabolites are diluted out of the system with constant dilution factor $\rho$. The magnification shows a schematic of the cellular sub-model, comprising five key processes: (1) uptake of metabolites, (2) their breakdown into other metabolites (Eqs 12 and 13), (3) generation of ATP through this process (Eq 8), and (4) the use of this ATP to drive protein synthesis (Eqs 5 and 6). (5) The cell proteome is partitioned into a constant housekeeping protein compartment ($Q$), a ribosome compartment ($R$), and a metabolic protein compartment ($P$). **B**: An example matrix of possible metabolic reactions. Metabolites can only react to form new metabolites that are one or two positions lower on the free-energy hierarchy. **C**: The thermodynamic trade-off that emerges from our model; ATP production rate increases linearly with an increase in ATP per reaction event ($\eta$), up to the point where thermodynamic inhibition becomes significant. The position of optimal $\eta$ value shifts as environmental metabolite concentrations change. **D**: The four possible species interaction types in our model. Competition and facilitation are present here like in all consumer-resource models that allow for the production of resources. In addition, there are two thermodynamic interaction types: "pollution" where one species produces a metabolite that causes thermodynamic inhibition of the other species, and "syntrophy" where a species consumes a metabolite that is thermodynamically inhibiting another species (and thus benefiting it).

where $\kappa$ is the substrate supply rate, $\delta_{\beta,\xi}$ is the Kronecker delta, $\rho$ is the metabolite dilution rate, $B$ is the number of species in the community, $p_{i,\beta}$ is the (per cell) rate that the $i^{\text{th}}$ species produces metabolite $\beta$, and $c_{i,\beta}$ is the (per cell) rate at which it consumes metabolite $\beta$. Further, $\lambda_i$ is the $i^{\text{th}}$ species' growth rate, $d_i$ its rate of biomass loss, $\tau_g$ is the characteristic time scale for growth, $\phi_R^*$ is the ideal ribosome fraction, $J_i$ is the ATP production rate for the $i^{\text{th}}$ species, $\chi$ is ATP use per elongation step, and $m$ is the total mass of the cell (in units of amino acids). The second term in Eq 4 corresponds to the energy use due to protein translation and the third term corresponds to the dilution of energy due to cell growth.

## Proteome partitioning

Due to finite cell size, a higher expression of a given type of protein must always come at the expense of lowered expression of other types of proteins. This means that a cell's ability to obtain energy for growth is constrained not just by thermodynamics, but also by the amount of protein it allocates to metabolism. In microbial models, this constraint is typically introduced in the form of a fixed "enzyme budget" that has to be divided between substrates [21]. However, the fact that the size of this "budget" varies with the amount of protein allocated to ribosomes has only recently begun to be considered [22], despite the fact that this constraint is

known to significantly impact microbial growth [12, 13]. Therefore, we develop a minimal cell model (Fig 1A), which extends mechanistic models of proteome constraints developed for homogeneous cell populations to the multi-species community level [14].

This sub-model comprises of five key processes: (1) The cell exchanges metabolites with its environment. For simplicity, we assume that the intracellular metabolite concentrations are equal to the environmental concentrations. Even if real cells can use a variety of mechanisms to maintain intracellular metabolite concentrations that make reactions more thermodynamically favourable, this assumption imposes the universally seen limit on efficiency that arises from the 2$^{nd}$ law of thermodynamics, i.e. maintaining a particular metabolite concentration necessarily consumes more free-energy than it contributes to the reaction free-energy. (2) Within the cell, substrates are broken down into lower free-energy metabolites. This process is thermodynamically-reversible in contrast to the conventionally used irreversible Michaelis–Menten kinetics (for more details see S1 Appendix). (3) This breakdown of substrates into lower free-energy metabolites allows free-energy to be transduced via the production of ATP. (4) This ATP is subsequently used to fuel protein synthesis [12–14]. Protein synthesis rate is thus dependent on ATP concentration, but sufficient free-energy is dissipated for it to be considered an irreversible process. (5) The proteome is partitioned into three compartments: a ribosome compartment $R$, a metabolic protein compartment $P$, and a compartment for all other proteins required by the cell $Q$ (termed "housekeeping"). As the housekeeping compartment is assumed to be constant, a direct trade-off between the fraction of the proteome dedicated to ribosomes and the fraction dedicated to metabolic proteins arises.

The growth rate of any given microbial species is determined by the total rate at which ribosomes synthesize proteins (process 4 in Fig 1A). This can be modelled by assigning each species ($i$) two internal variables, the internal energy (ATP) concentration $a_i$ and the ribosome fraction $\phi_{R,i}$. The growth rate depends on both quantities and can be expressed as

$$\lambda_i(a_i, \phi_{R,i}) = \frac{\gamma(a_i) f_b \phi_{R,i}}{n_R}, \tag{5}$$

where $\gamma(a_i)$ is the effective translation elongation rate, $f_b$ is the average fraction of ribosomes bound and translating, and $n_R$ is the number of amino acid per ribosome. As we assume that protein synthesis is an irreversible process, the effective translation elongation rate is assumed to be saturating with respect to the energy concentration, taking the form

$$\gamma(a_i) = \frac{\gamma_m a_i}{\gamma_{\frac{1}{2}} + a_i}, \tag{6}$$

where $\gamma_m$ is the maximum elongation rate, and $\gamma_{\frac{1}{2}}$ is its half-maximum constant. With Eqs 5 and 6, we can now define the population dynamics in Eq 2 in terms of the cells' internal state. We therefore now consider the dynamics of a cell's internal state, starting with the ribosome fraction dynamics. We assume that for a specific internal energy concentration ($a_i$) each cell aims to reach a particular ribosome fraction $\phi_R^*(a_i)$ (the "ideal" ribosome fraction). Similar to Eq 6 it is assumed to be a saturating function of energy concentration,

$$\phi_R^* = \frac{a_i}{\Omega_{\frac{1}{2}} + a_i}(1 - \phi_Q), \tag{7}$$

where $\Omega_{\frac{1}{2}}$ is a half-maximum constant and $\phi_Q$ is the housekeeping protein fraction. By also noting that the characteristic time scale for growth is given by $\tau_g = \log_2(100)/\lambda_i$ (see S1 Appendix),

ribosome dynamics in Eq 3 can now be solved for a given value of the internal energy concentration $a_i$.

## ATP production

To determine the dynamics of the cellular internal energy concentration the rate at which each species produces ATP (process 3 in Fig 1A) must be determined. Fig 1B shows the general pattern of possible reactions in our model for an example case with a small number of metabolites ($M = 5$). Due to the explicit thermodynamic reversibility in our model, only reactions descending in free-energy are allowed. Further to this, the enzyme scheme we used is intended to capture simple enzymatic reactions rather than complex reactions composed of a succession (e.g. glucose respiration). Hence, in our model direct links are only allowed between metabolites with small separations (two positions or less) on the metabolite hierarchy. This network of reactions is fully connected, i.e. every metabolite can be reached from any metabolite higher in the metabolite hierarchy. Each species ($i$) is assigned a random subset ($O_i$) of this set of possible reactions. As this subset is small, the reaction networks of individual strains are typically incomplete, although for sufficiently complex communities the community-level reaction network would tend to be fully connected. A set amount of ATP ($\eta_{\alpha,i}$) is generated for each species for each reaction $\alpha$. The ATP production rate can then be found by summing the product of this and the reaction rate as

$$J_i = \sum_{\alpha \in O_i} \eta_{\alpha,i} q_{\alpha,i}(E_{\alpha,i}, S_\alpha, W_\alpha),$$

(8)

where $q_{\alpha,i}$ is the $i$th species' (mass specific) reaction rate for reaction $\alpha$, $E_{\alpha,i}$ is the copy number of enzymes for reaction $\alpha$ that the species possesses, $S_\alpha$ is the concentration of reaction $\alpha$'s substrate, and $W_\alpha$ is the concentration of of reaction $\alpha$'s waste product. The enzyme copy number depends on metabolic protein fraction, and so can be expressed in terms of the ribosome fraction as

$$E_{\alpha,i} = \frac{m v_{\alpha,i} \phi_{P,i}}{n_P} = \frac{m v_{\alpha,i}(1 - \phi_{R,i} - \phi_Q)}{n_P},$$

(9)

where $v_{\alpha,i}$ is the $i$th species' proportional expression level for reaction $\alpha$, and satisfies $\sum_{\alpha \in O_i} v_{\alpha,i} = 1$, $\phi_{P,i}$ its metabolic protein fraction, and $n_P$ is the mass (average number of amino acids) of a metabolic protein [14]. The reaction rate in Eq 8 is derived by assuming a reversible kinetic scheme (derivation in S1 Appendix) and can be expressed as

$$q_{\alpha,i}(E_{\alpha,i}, S_\alpha, W_\alpha) = \frac{k_{\alpha,i} E_{\alpha,i} S_\alpha (1 - \theta_i(S_\alpha, W_\alpha))}{K_{S_\alpha,i} + S_\alpha(1 + r_{\alpha,i}\theta_i(S_\alpha, W_\alpha))},$$

(10)

where $k_{\alpha,i}$ is the maximum forward rate for reaction $\alpha$ for the $i$th species, $K_{S_\alpha,i}$ is its substrate half-saturation constant for reaction $\alpha$, $r_{\alpha,i}$ is its reversibility factor for reaction $\alpha$, and $\theta_i(S_\alpha, W_\alpha)$ is a thermodynamic factor given by

$$\theta_i(S_\alpha, W_\alpha) = \frac{Q(S_\alpha, W_\alpha)}{\mathcal{K}_{\alpha,i}},$$

(11)

where $Q(S_\alpha, W_\alpha)$ is the reaction quotient, which in the single-reactant/single-product case we consider, is defined as $Q(S_\alpha, W_\alpha) = W_\alpha/S_\alpha$, and $\mathcal{K}_{\alpha,i}$ is the $i$th species's equilibrium constant for reaction $\alpha$. The thermodynamic factor $\theta_i(S_\alpha, W_\alpha)$ quantifies how far from equilibrium the reaction is, taking a value of 1 at equilibrium and tending towards 0 far from equilibrium. It is

worth noting that for high $\mathcal{K}_{\alpha,i}$ values Eq 10 will not significantly differ from the Michaelis–Menten case. With the rate of energy acquisition ($J_i$) now sufficiently defined, the dynamics of the internal energy concentration ($a_i$) can be obtained by Eq 4. Finally, we wish to consider the impact the species have on the environmental metabolite concentrations through the production and consumption of metabolites (process 2 in Fig 1A). From the expression for the reaction rate, the consumption and production rates for metabolite $\beta$ that appear in Eq 1 can be defined as

$$p_{i,\beta}(\mathbf{C}) = \sum_{\alpha \in O_i} \delta_{C_\beta, W_\alpha} q_{\alpha,i}(E_{\alpha,i}, S_\alpha, W_\alpha),$$

(12)

and

$$c_{i,\beta}(\mathbf{C}) = \sum_{\alpha \in O_i} \delta_{C_\beta, S_\alpha} q_{\alpha,i}(E_{\alpha,i}, S_\alpha, W_\alpha).$$

(13)

In the above, $\delta_{C_\beta, W_\alpha}$ and $\delta_{C_\beta, S_\alpha}$ are Kronecker deltas that are zero unless metabolite $\beta$ is the waste product or substrate of reaction $\alpha$, respectively.

## Free-energy dissipation and thermodynamic inhibition

Due to the thermodynamic reversibility in our model, the amount of free-energy dissipated affects reaction rates. We express the amount of free-energy obtained from a given reaction event as the parameter $\eta$ (units of number of ATP molecules). Cells can also transduce free-energy by pumping ions across membranes. Hence, we assume the minimum value this parameter can take is $\eta = 1/3$, which is approximately equivalent to the amount of free-energy transduced by pumping one ion across a membrane. The maximum possible $\eta$ value corresponds to all of the free-energy being transduced to ATP and none of it being dissipated. However, in this case the overall reaction will be at equilibrium and there will be no net production of ATP. This impact of free-energy dissipation on the dynamics occurs through the equilibrium constant which is specified as

$$\mathcal{K}_{\alpha,i} = \exp\left(\frac{-\Delta_\alpha G^0 - \eta_{\alpha,i} \Delta G_{\mathrm{ATP}}}{RT}\right),$$

where $T$ is temperature, $R$ is the gas constant, $\Delta G_{\mathrm{ATP}}$ is the Gibbs free-energy per mole of ATP, and $\Delta_\alpha G^0$ is the standard Gibbs free-energy change when one mole of reaction $\alpha$ occurs. As $\eta$ changes, the reaction rate (Eq 10) changes due to the change in this equilibrium constant. An example of the impact this has on the ATP production rate is visualised in Fig 1C. For low $\eta$ values, the equilibrium constant is very large so there is a negligible thermodynamic impact on the dynamics, and thus the ATP production rate initially scales linearly with $\eta$. However, the exponential form of the equilibrium constant means that there is only a narrow region with anything other than negligible or complete thermodynamic inhibition. This means that ATP production peaks and then rapidly declines to zero as $\eta$ increases. The narrowness of this rate-yield trade-off means that there exists a clear optimal strategy with alternative thermodynamic strategies being sub-optimal. However, as the position of the peak of this trade-off is determined by the waste product concentration, environmental conditions will determine the optimal strategy.

The inclusion of thermodynamic reversibility in our model has the additional benefit of allowing the entropy production rate (free-energy dissipation rate) to be directly calculated. To do this the free-energy dissipated per reaction event of reaction $\alpha$ must be found for each

species ($i$). This can be expressed as

$$D_{i,\alpha} = \Delta_\alpha G^0 + RT \ln(Q(S_\alpha, W_\alpha)) + \eta_{\alpha,i}\Delta G_{ATP}.$$

From this, whole community entropy production rate is then found to be

$$\frac{dG_d}{dt} = \sum_{i=1}^{B}\left(\sum_{\alpha \in O_i}\left[\frac{D_{i,\alpha}q_{\alpha,i}(E_{\alpha,i}, S_\alpha, W_\alpha)}{T}\right]N_i\right). \tag{14}$$

This entropy production can be used as a state variable allowing the complex dynamics involved in microbial community assembly to be tracked in a simpler but still physically meaningful manner. Without the overall summation the entropy production of individual species can also be found.

## Emergence of novel interaction types

The types of species interactions that emerge from this model and the mechanisms that lead to them are illustrated in Fig 1D. Similarly to other microbial consumer-resource models [23–25], species in our model can interact by competing for shared substrates (competition) or by one species producing a metabolite that the other uses (facilitation). In addition, due to the possibility of thermodynamic inhibition, interactions via waste products (rather than substrates) are now possible. Furthermore, species can also now interact by one species producing a product that thermodynamically inhibits the other species (pollution), or by one species consuming a product that inhibits the other (syntrophy). Example plots for syntrophy and pollution interactions (in a simple community) are shown in S1 and S2 Figs, respectively. For the more complex communities we move on to consider, these interactions are determined by perturbing each metabolite in turn at steady state, and calculating the response of each species. Then, the net impact of each species on the concentration of each metabolite is calculated, which combined with the perturbation response allows interaction strength between each species for each metabolite to be quantified. This method is reminiscent of the calculation of "susceptibilities" in the cavity method [26, 27]. An important difference is that our responses are calculated numerically at steady state, as analytic "susceptibilities" could not be obtained. This was due to reaction rates in our model depending on both substrate and waste-product concentrations, which entails greater dynamically coupling between different metabolites. Finally, the interaction types are classified using the sign of the perturbation response and whether the species makes a net positive or negative impact on the concentration of the perturbed metabolite (for the full process see S1 Appendix).

## Simulations

To simulate community assembly we numerically integrated the system of $M + 3B$ ordinary differential equations (Eqs 1–4). We generated the ($M$ metabolite) reaction networks following the pattern shown in Fig 1B, with the free-energy spacing of each step downwards in the metabolite hierarchy being equal. For each community, we then generated a random set of $B$ species. Each species was assigned a number of reactions ($N_O$) drawn randomly from a uniform distribution. We then generated the reaction set ($O$) for that species by drawing the $N_O$ reactions at random from the full reaction network. We assigned random kinetic parameters ($k$, $K_S$ and $r$) to each reaction of each species, drawn from log-normal distributions as they must be strictly positive. Subsequently, we set the relative reaction expression levels ($v$) by assigning each reaction a uniformly distributed random number normalised to the sum of all reactions for the species in question. The next step was to assign the $\eta$ values to each reaction

for a species. When $\eta$ values are sufficiently low, the dynamics remain far from equilibrium and the enzyme kinetics are effectively irreversible Michaelis–Menten. Thus, we can generate communities with irreversible Michaelis–Menten enzyme kinetics simply by restricting the maximum $\eta$ value. Henceforth, for the sake of brevity we refer to these communities as having Michaelis–Menten enzyme kinetics. We randomly chose $\eta$ values from a uniform range between the minimum value ($\eta = 1/3$) and a maximum value which varies based on whether we were considering Michaelis–Menten or reversible kinetics. For Michaelis–Menten kinetics, the maximum possible value of $\eta$ was chosen such that the reaction will reach equilibrium at a product to substrate ratio of $1 \times 10^5$. If instead we considered reversible kinetics, the maximum possible value of $\eta$ was chosen such that the reaction will reach equilibrium at a product to substrate ratio of $1 \times 10^{-2}$. After we had generated the random communities, each replicated assembly simulation was initiated with equal abundance across species and the system numerically integrated till the dynamics reached steady state ($1 \times 10^8$ seconds). A table of parameters and parameter ranges we used for this model can be found in S1 Appendix.

## Results

The dynamical behaviour of our model is demonstrated in Fig 2. The population dynamics (Eq 2) of the species that survive to steady state, along with a small number of species that go extinct, are shown in Fig 2A. The corresponding ribosome fraction dynamics (Eq 3) are shown in Fig 2B. The fractions increase during favourable conditions, leading to an increased growth rate before decreasing to a steady state value (for surviving species). All species are observed to converge on the same ribosome fraction at steady state. This occurs because the values of the two half-maximum constants ($\gamma_{\frac{1}{2}}, \Omega_{\frac{1}{2}}$) are fixed across species, meaning a decreased ribosome fraction (Eq 7) cannot be counteracted by an increased translation rate (Eq 6). In Fig 2C the metabolite concentration dynamics (Eq 1) for this community are shown. Initially, the single supplied metabolite accumulates, but as the population of species using this metabolite increases its concentration decreases and the concentration of metabolites one or two steps down the hierarchy increases. This process repeats leading to a sequential diversification of substrates, which leads to clear shifts seen in the ribosome fraction and population dynamics. In Fig 2D the rate of community entropy production (Eq 14) is plotted. The entropy production rises as the substrate diversifies and the total population increases, and shows clear peaks at the time points where accumulated substrates are rapidly depleted. The number of entropy production spikes shows a strong correlation with the final number of substrates diversified (Fig 2D inset).

### Final community states depend on free-energy availability

We now assign every microbe to a functional group based on the substrate of its most expressed reaction. The relative abundance of these functional groups with time is shown for a representative parameter set in Fig 3A. The functional diversity initially collapses and then slowly relaxes towards steady state. Using the inverse Simpson index, which corresponds to the effective number of types (here functional groups) in an community we find that functional diversity collapse is common across our simulations. The final community states are now compared across four different regimes in Fig 3B. The regimes compared are low free-energy (recalcitrant) substrates and high free-energy (labile) substrates, and whether the enzyme kinetics are Michaelis–Menten or reversible. For all regimes, species are assigned between 1 and 5 reactions from a 25 metabolite (47 reaction) network, and simulations are started with 250 species. The first property compared is the number of surviving functional groups. Systems supplied with a substrate of higher free-energy see a greater number of

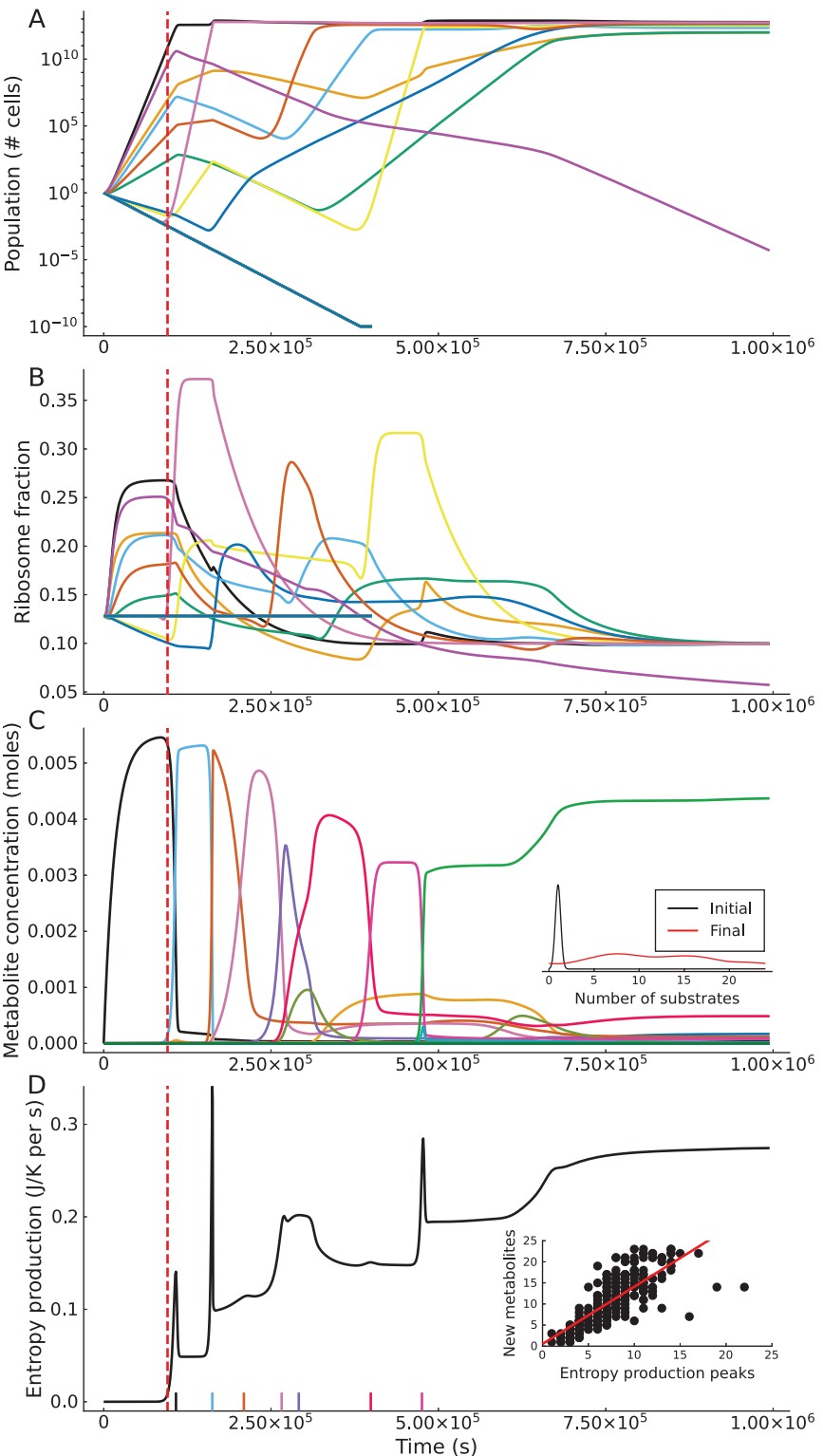

**Fig 2. Key dynamical behaviours of the model.** The vertical dashed lines in all plots mark the time point where the third metabolite is available in sufficient quantity ($1 \times 10^{-4}$ moles) to support growth. This specific simulation is started with an initial community of 250 species, each of which is assigned between 1 and 5 reactions from a 25 metabolite (47 reaction) network. The parameter set used here was one of the 250 parameter sets generated using a reversible kinetic scheme and a total free-energy change of $5.0 \times 10^6$ J mol$^{-1}$. **A:** Microbial population dynamics. **B:** Ribosome fraction

dynamics. Note that all species that survive to -steady state settle to the same ribosome fraction that balances the constant biomass loss rate. **C**: Metabolite dynamics. In contrast to **A**, these are shown on a linear scale to show the changes in key metabolite concentrations. The inset shows densities of the initial and final number of metabolites across 250 simulations. **D**: Entropy production rate of the community with time. The time points where accumulated metabolites drop below an exhaustion threshold ($2 \times 10^{-3}$ moles) are marked on the x-axis (colour-coding corresponds to plot **C**). In the inset the number of entropy production spikes is plotted against the number of new substrates generated (over 250 simulations) with a correlation of 0.769. The best fit line (red) shows a slope of 1.36 and a y intercept of 0.523.

surviving functional groups. Using a reversible kinetic scheme (rather than Michaelis–Menten) only increases the number of surviving functional groups in the low free-energy case. A nearly identical pattern is observed for the number of surviving species, suggesting that the decline in functional diversity is predominantly driven by a decline in species diversity. To test whether high free-energy substrates can support more species we then compared the ratio between the number of surviving species and the number of substrates diversified to. We found that the number of survivors per substrate is lower for high free-energy substrates, with no significant difference between kinetic schemes. For the community entropy production rate, higher substrate free-energy corresponds to higher entropy production rates. There is again only a significant difference between kinetic schemes when substrate free-energy is low.

## Free-energy availability increases rate of niche generation

The naive explanation for the diversity results in Fig 3B is that higher free-energy availability increases the chance of a species being viable on a particular substrate. However, this is clearly contradicted by the surviving species per substrate results shown in Fig 3B. A more careful investigation of the mechanism that sustains diversity is displayed in Fig 4. The first variable we consider is the number of surviving species, which is shown in Fig 4A. All regimes show a significant loss in species diversity at a specific time. This is the time point where species that never grow at all reach extinction. The number of surviving species drops to a significantly lower value across all regimes and then levels out, remaining at a substantially higher level for high free-energy (labile) substrates. When low free-energy (recalcitrant) substrates are

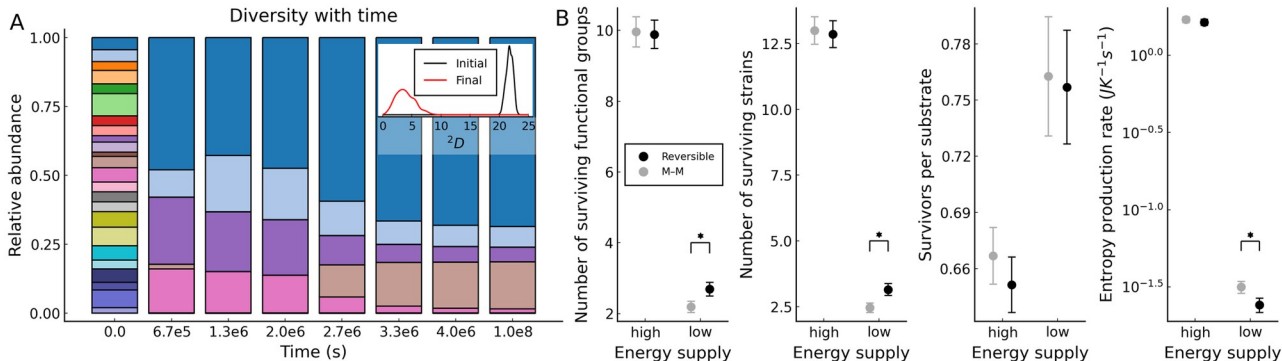

**Fig 3. Free-energy availability increases species diversity. A**: Relative abundances of functional groups over time. The inset density plot shows initial and final distributions of inverse Simpson's diversity indices for functional groups across 250 simulations (using the same simulation parameters as Fig 2). **B**: Comparisons between the four free-energy × reversibility regimes. For each regime, the average and 99% confidence intervals obtained from 250 simulations are plotted. Two values of the total free-energy of the supplied substrate are considered: high ($1.5 \times 10^7$ J mol$^{-1}$), and low ($1.5 \times 10^6$ J mol$^{-1}$). For both energy regimes, we consider both reversible (black symbols) and Michaelis–Menten (M–M grey symbols) kinetics. Cases with a significant difference between these pairs are marked with a star ($P < 0.01$). The greatest differences are seen between high (labile) and low (recalcitrant) free-energy substrates, and only in cases of low free-energy does the kinetic scheme used cause significant differences.

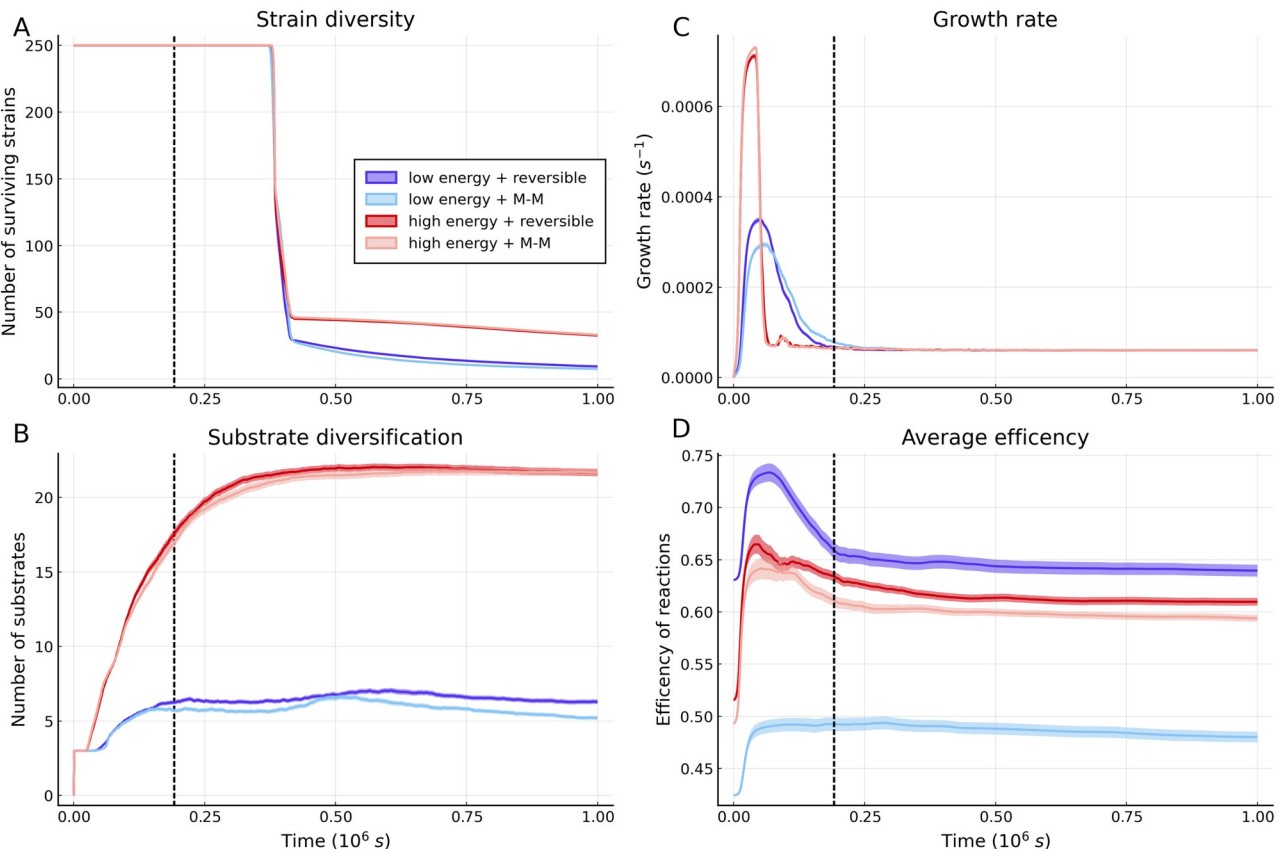

**Fig 4. Averaged assembly patterns across communities.** Averages (across 250 simulations) of key quantities over time are shown (with 99% confidence intervals) for the four regimes considered in Fig 3. To better reveal the dynamics only a small initial time window is shown. **A**: Average species diversity over time. The dashed black line shown in all four plots marks the time point where 75% of species have dropped below the threshold where they can no longer significantly impact the metabolite dynamics. **B**: Substrate diversification plateaus for all regimes at approximately this time. **C**: Across all regime growth rate rapidly peaks, the peak is both earlier and higher for high free-energy (labile) substrates. **D**: Dynamics of the average reaction efficiency, i.e. the fraction of the free energy change that is used to generate ATP rather than being dissipated (full definition provided in S1 Appendix). The position of the peaks can be seen to match the position of the corresponding growth rate peaks in plot **C**. The four plots use the same parameters for the respective regimes as are used in Fig 3.

considered, a small but significant difference in the number of survivors between the reversible and Michaelis–Menten regimes can be seen to emerge after the mass extinction event.

The second variable we consider is the number of substrates diversified, shown in Fig 4B. For all regimes, the number of substrates rapidly plateaus. The plateau occurs after the point where the majority of species can no longer contribute to the substrate dynamics. At high substrate free-energy, the plateau occurs at a far higher number of substrates than at low substrate free-energy, and at low substrate free-energy the reversible case plateaus slightly higher than the corresponding Michaelis–Menten case. This mirrors the species diversity shown in Fig 4A, indicating that diversity is driven by the number of available substrates (niches) being generated during assembly. The average growth rates for the four regimes are shown in Fig 4C. All regimes show an initial period of rapid growth while the energy availability per species is high which settles down to a steady growth rate as the total population increases. The most pronounced peaks are seen for high free-energy (labile) substrates due to the greatly increased energy availability. There is a small difference in peak height for low free-energy (recalcitrant) substrates as allowing reactions close to equilibrium leads to greater free-energy extraction

from the environment. The vastly different growth rates across species' populations account for the differing rates of niche generation (substrate diversification) seen in Fig 4B.

The strategy of allowing near-equilibrium reactions increases free-energy yield, but also means that thermodynamic equilibrium occurs at far lower waste product concentrations. This means that the chance of a species expressing protein for a reaction that cannot proceed becomes far higher. To ascertain the importance of this effect we plot average reaction efficiencies (defined in S1 Appendix) with time in Fig 4D. The average thermodynamic efficiency of the community changes through a process of species sorting, as the proportional abundance of species with differing average reaction efficiencies changes. Species sorting drives an increase in average reaction efficiency during the initial growth period across all regimes, as species with highly efficient reactions yield more free-energy, and thus grow faster. The two low free-energy (recalcitrant) cases are substantially further apart than the two high free-energy (labile) cases. This arises because low free-energy reactions are inherently closer to equilibrium. Hence, the reversible case has higher efficiency because its closer to equilibrium, and the efficiency of the Michaelis–Menten case is reduced more significantly to ensure reactions are far from equilibrium. In the reversible low free-energy case, after the initial period waste products accumulate and reactions become thermodynamically inhibited, a sharp efficiency peak is created. In the corresponding Michaelis–Menten case, the reactions are always far-from-equilibrium and so this inhibition never occurs and the efficiency simply plateaus. As the higher free-energy (labile) cases are generally further from equilibrium, the pattern here is likely driven by a transition from an initial highly efficient community that breaks down the initial substrate to more diverse community that breaks down a wider range of secondary substrates. This community is less thermodynamically efficient as the broader range of substrate means that competition for metabolites (ultimately, free-energy) is weaker.

## Thermodynamic interaction types influence community assembly and diversity

Finally, we examined the preponderance of the different species interaction types (cf. Fig 1D) under the low/high free-energy and reversible/irreversible regimes. As the choice of kinetic scheme only significantly affected previous results for low free-energy (recalcitrant) substrates, we first compare kinetic schemes for this case in Fig 5. We show that under Michaelis–Menten like enzyme kinetics (Fig 5A), there is a clear separation of scales between the two conventional (competition and facilitation) interaction types on the one hand, and the two thermodynamic ones (syntrophy and pollution) on the other. Our results thus imply that when considering the Michaelis–Menten (high dissipation) limit of a reversible kinetic scheme, thermodynamic interactions are observed (Fig 5B), but are so weak as to not affect the dynamics (Fig 5A). In contrast, when considering a reversible kinetic scheme the separation of scales between the strengths of the thermodynamic and non-thermodynamic interaction types disappears (Fig 5C), and thermodynamic interactions become marginally more common (Fig 5D). Taken together, this implies that thermodynamic interactions have a meaningful impact on the overall community dynamics when a reversible kinetic scheme is used and substrate free-energy is low. In S3 Fig the results are shown for the high free-energy (labile) case, where a similar overall pattern is seen. However, the proportion of thermodynamic interactions is significantly lower, as is the mean strength of the thermodynamic interactions relative to the non-thermodynamic interactions. This is expected because the kinetic scheme is largely irrelevant to interaction dynamics when the system is assembled on a high free-energy substrate.

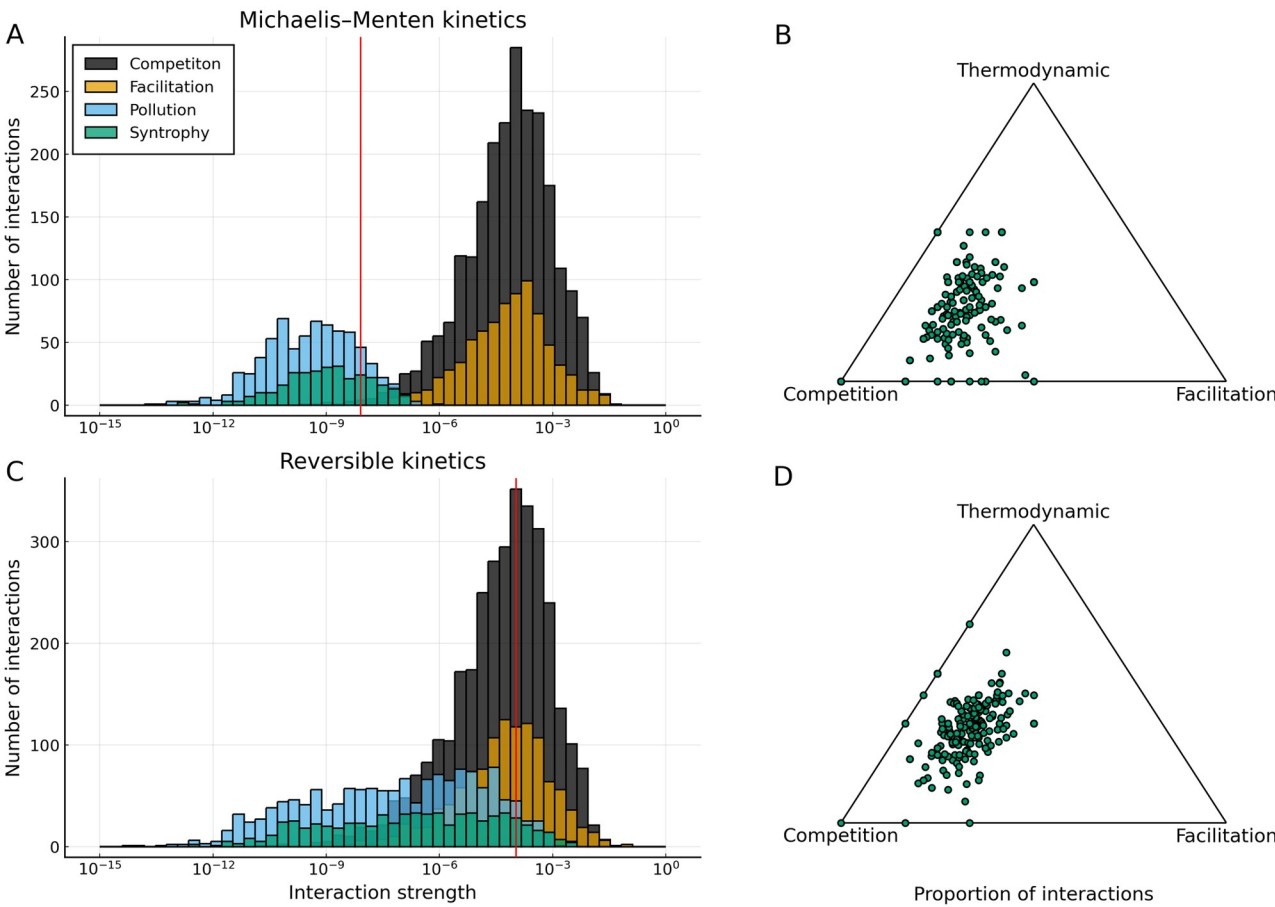

**Fig 5. Thermodynamic and conventional interactions can be of comparable strength. A**: Histogram of strengths of the four interaction types (shown schematically in Fig 1D). The vertical red line marks the point where the distribution of the strengths of the pollution interaction is at half its maximum. A clear separation of strengths between conventional and thermodynamic interactions can be observed. **B**: Simplex showing the relative proportions of competition, facilitation and the thermodynamic (pollution and syntrophy) interaction types. Each point represents one out of the 250 communities we simulated for each regime. Competition is by far the most common interaction type, with thermodynamic and facilitation interactions being similarly common. **C**: Corresponding plot to Fig 5A for reversible kinetics. Thermodynamic and conventional interactions now reach comparable strengths. **D**: Corresponding plot to panel B for reversible kinetics. Competition remains the most common interaction type, but thermodynamic interactions are now marginally more common than facilitation. All cases considered here are for low free-energy (recalcitrant) substrates ($1.5 \times 10^6$ J mol$^{-1}$).

## Discussion

We have shown that differences in the diversity of microbial communities are strongly influenced by the free-energy level of the substrates that they are assembled on (Fig 3). This is because high free-energy (labile) substrates allow more rapid creation of niches during the initial phase of assembly. At the end of this initial phase of assembly, diversity collapses because species that are not able to grow due to the lack of available substrates (niches) go extinct. As more niches are generated in the high-free energy case than in the low-free energy case, a greater level of species diversity to be sustained at steady state (Fig 4). We also find that for low free-energy (recalcitrant) substrates near-to-equilibrium reactions significantly increase the availability of free-energy, and thus the diversity. However, after the initial phase of community establishment, they are disfavoured as waste products build up. Underlying these dynamics are distributions of the different types of species interactions: thermodynamic interaction types (pollution and syntrophy) are rarer than the conventional ones (competition and

facilitation), but they can reach similar magnitudes of strength when the supplied substrate has low free-energy and the kinetics are reversible (allowing non-equilibrium thermodynamics to operate) (Fig 5). Taken together, these results indicate that thermodynamic inhibition can constrain the diversity of microbial communities when free-energy is scarce.

Thermodynamic constraints are known to be generally more important in anaerobic than aerobic microbial communities due to the lower free-energy availability. This is why previous microbial community models incorporating thermodynamic inhibition were developed specifically for anaerobic systems [15, 16]. However, aerobic microbial communities frequently assemble on relatively recalcitrant substrates in nature (e.g., high-cellulose or high-chitin resources; [28]). Therefore our model and results are relevant beyond anaerobic systems. For the relatively labile substrates typically used in laboratory experiments, a non-thermodynamic model would be adequate. However, when modelling growth on more realistic recalcitrant substrates thermodynamic effects are likely to be important. Most microbes preferentially use labile substrates, but shifts towards using recalcitrant substrates has been observed to occur both as temperature increases [29], and as communities assemble [30]. Thus, understanding community-level thermodynamic constraints when the substrate is recalcitrant is necessary for truly understanding microbial community assembly in nature.

The relative importance of competitive versus cooperative interactions to microbial communities is currently an open question. Most of this work has taken a phenomenological empirical approach to characterise and classify such interactions [31]. Whole-genome community-scale metabolic modelling now allow a more mechanistic understanding of these interactions by predicting both the substrate overlap and metabolite production (leading to facilitation) between microbial species [32]. However, these approaches necessarily assume steady state metabolite concentrations, and do not account for the nonlinear feedback loops that develop in dynamic complex real microbial communities. We therefore took the middle road in terms of model complexity by incorporating non-equilibrium thermodynamics into a conventional consumer-resource model with cross feeding [19, 23]. Doing this introduces a dependence of reaction rate on product concentration, which leads to the emergence of novel interaction types between species via products rather than via substrates (see Figs 1D and 5). Our results suggest that characterizing communities simply by either the relative proportion of positive and negative interactions, or the relative strength of competition and facilitation is insufficient to capture the true complexity of microbial interactions and their effects on community dynamics.

Because thermodynamic reversibility is explicitly included in our model, we are able to directly calculate the rate at which communities produce entropy (Eq 14). This is a key advance because despite the long-standing interest in ecology about entropy production [2, 3], calculation of entropy production rates from ecological models and ecosystems has not been done in a explicit and consistent manner. Fig 2 shows that spikes in the rate of entropy production over time can indicate periods where the availability of resources are changing rapidly. This has potential empirical relevance as it suggests that key episodes in the development of microbial communities could be detected by the rate at which they add heat to their environment. Additionally, as seen in Fig 3B, the rate of entropy production at steady state can be an indicator of free-energy availability, which only varies significantly between kinetic schemes in the case of low free-energy (recalcitrant) substrates. Potentially rates of community entropy production calculated from our model could be compared with experimental measurements of heat production, in order to establish the thermodynamic efficiency of microbial communities.

A comparison of our models assumptions and results with other recent work can be found in S1 Appendix. To the best of our knowledge no previous work has combined both proteomic

and thermodynamic constraints into a microbial community assembly model. Some of the previous studies find, in contrast to our results, that the number of surviving consumer species can exceed the number of substrates. This generally stems from the existence of distinct metabolic strategies, i.e. differences between species in their allocation of enzymes to different substrates. Entire communities are found to be able to survive if the resource supply vector is contained by the convex hull formed by the community's set of strategies [21, 33]. This enhanced community diversity can be further facilitated by adaptive proteome reallocation between reactions [22]. Reconciling our model's results with these previous models would require the development and analysis of a unified model, combining thermodynamic constraints with adaptive reallocation of proteome between reactions. In addition, a recent study by Marsland et al. also found a dependence of final species diversity on energy supply [23]. Our model offers additional insight, by establishing a clear link between free-energy availability and the dynamics of microbial community assembly. This insight was only possible because our model incorporated both proteomic and (non-equilibrium) thermodynamic constraints.

A major assumption of our model is that the metabolite dynamics of the whole community can be captured by merely considering well mixed external metabolites that are accessible to all species, i.e. ignoring local metabolite concentrations and internal metabolites. This assumption is frequently made in microbial consumer-resource modelling [23, 24, 34]. If we were to include internal metabolites, species would be able to maintain concentration gradients (through uptake and excretion), thus preventing reactions from reaching thermodynamic equilibrium. It is important to note however, that for the catabolic reactions that we consider, the free-energy cost of maintaining a particular concentration gradient will always, due to the $2^{nd}$ law of thermodynamics, be greater than the reaction free-energy contributed by the gradient. The limits that thermodynamic inhibition place on microbial growth cannot therefore be evaded through this mechanism. Another limitation of our study is that we only considered metabolite hierarchies with equal free-energy spacing between metabolites. Extending this to allow free-energy gaps of variable sizes in the same hierarchy would mean that thermodynamic bottlenecks develop, which mean that species need to invest substantial amounts of protein in reactions close to thermodynamic equilibrium in order for the system to fully develop [35]. This potentially allows the thermodynamics of more realistic scenarios, such as overflow metabolism to be investigated [36, 37].

Recently, the existence of an (organism specific) upper limit on the rate of free-energy dissipation (entropy production) has been suggested [38]. Niebel et al. consider cells that exhibit "overflow metabolism", i.e. incomplete oxidisation of their growth substrate. By increasing its use of metabolic overflow pathways, the cell makes increased use of reactions that dissipate less free-energy relative to the free-energy retained for growth, thus resulting in a maximum free-energy dissipation rate. Cells cannot adapt their metabolism in such an manner in our model, where changes in the community-level distribution of metabolic strategies changes solely through species sorting. However, in our model a maximum metabolic flux is effectively imposed because both the metabolic proteome fraction ($\phi_P$) and the (randomly chosen) kinetic parameters are bounded. Thus, given that every substrate has a fixed Gibbs free-energy, a specific maximum possible free-energy dissipation rate is implied by this maximum metabolic flux. We would not expect this dissipation rate to be reached due to competition between species, and would instead expect the system to settle to a lower dissipation rate (see Fig 2B). Interestingly, our maximum dissipation rate arises from proteomic rather than thermodynamic constraints, suggesting a potential link with proteomic constraint-based explanations for the existence of overflow metabolism [36]. It is worth noting that if cells are genuinely constrained by the need to remove the entropy generated by free-energy dissipation, then thermodynamically efficient reactions would be additionally favoured as they produce less entropy.

Species would therefore be expected to operate closer to thermodynamic equilibrium, implying that thermodynamic interactions could be even more important than our results suggest.

Our results provide a new perspective on the apparent simplicity and modularity of assembly dynamics reported by recent empirical studies [24, 39, 40]. In Goldford et al., a diverse initial community was inoculated onto a single carbon source [24], and the relative abundances of various taxa tracked over successive dilutions. Similar to Fig 3A they observed a rapid collapse in diversity, which then levelled out to a relatively simple community with cross-feeding. Our results highlight that this pattern would very likely be impacted by substrate lability. Datta et al. considered assembly in an open system (one where new species can enter) [39], finding a highly reproducible assembly process consisting of multiple distinct phases, throughout which species diversity was observed to vary non-monotonically. In contrast, in our closed system diversity declines monotonically, and thus the rate of niche generation is of critical importance (see Fig 4C). Generated niches cannot contribute to diversity after the species that could feasibly occupy them have gone extinct. Furthermore, consistent with both real systems [24, 40] and conventional microbial consumer-resource models [34], modularity emerges in our model microbial communities (see Fig 3A). We observe that the first functional group predominates across all regimes, but the relative abundances of the secondary functional groups are highly contingent as new species cannot enter the system, so the large number of niches generated by substrate diversification are not guaranteed to be filled. Functional groups also possess definite patterns of interaction types (see Figs 1D and 5) with competition occurring primarily within functional groups, facilitation and syntrophy occurring primarily between functional groups, and pollution occurring both within and between neighbouring functional groups (for further discussion of this point see S1 Appendix).

Our model was designed to have sufficient complexity to capture the effects of cellular proteome fractions and non-equilibrium thermodynamics, which are generally ignored in community-scale models. Modelling at the intermediate scale provides novel, empirically-relevant insights, while avoiding the high complexity of models with fully-explicit intra-cellular dynamics [41]. Including a proteome trade-off generates a direct mechanistic link between free-energy availability and growth rate, because in regimes of greater free-energy availability, less of the proteome needs to be dedicated to metabolism, leading to a higher ribosome fraction, and thus the higher maximum growth rates seen in Fig 4C. One of the ways that we reduced model complexity, was by making the simplifying assumption that the saturation constants, for both translation rate and proteome fraction, were fixed across species. Relaxing this assumption would allow different ribosome expression strategies to coexist in our model, leading to meaningful variation in steady-state ribosome fraction. Fruitful investigation into the role that different proteome strategies play in microbial community assembly would then be possible, particularly in regards to the observed negative relationship between the number of ribosomal RNA operons a species possesses and its carbon use efficiency [42]. We were also unable to explicitly test whether a specialist vs generalist trade-off exists in our model, because the substrate supply conditions we studied systematically favoured species with more reactions, as they had a greater probability of possessing a reaction for utilizing available substrates early in the assembly process. Changing the assembly process to allow continual immigration of species into the system would allow proper investigation into the existence and strength of the specialist vs generalist trade-off. Our framework could also fruitfully be extended by making links with metabolic ecology, particularly with the suggestion that the recently-reported variation in thermal sensitivity of microbial growth rates [43, 44] can be linked to proteome allocation [45].

In summary, our results show that an explicitly thermodynamic model of complex, dynamically assembling ecosystems can provide novel insights into the mechanisms that generate

and maintain diversity, how diversity depends on free-energy availability, the role of entropy production, and the nature of underlying interactions between populations. This illustrates the value and importance of considering non-equilibrium thermodynamics explicitly in the modelling of microbial communities.

## Supporting information

**S1 Appendix. Supplementary text.** A comparison of Michaelis–Menten and reversible enzyme kinetic schemes, a full derivation and validation of the proteome trade-off model, a definition of the measure used to quantify reaction efficiency, details of the method used for identifying interactions and quantifying their strengths, a comparison of our results with those of previous models, heat-maps showing the strength and frequency of the different interaction types between functional groups, tables of model parameters, a table summarising the assumptions underlying our model, and plots demonstrating the robustness of our results to changes in the parameterisation.
(PDF)

**S1 Fig. Syntrophy leads to favouring more efficient species.** Our system here consists of three metabolites (only the first of which is supplied) and three species. The species A and B both break down metabolite 1 to produce metabolite 2, and species C breaks down metabolite 2 to produce metabolite 3. Species B generates more ATP per mole of reaction than A does. **A**: When species A and species B are grown together species A initially grows faster due to its greater ATP yield. However, as steady state is approached species A dies off to be replaced by species B. **B**: Species A dying off occurs due to the build of metabolite 2, which inhibits species A more due to it being closer to thermodynamic equilibrium. **C**: When species C is included, species A now survives to steady state (along with species C) instead of species B. **D**: The concentration of metabolite 2 is reduced due to consumption by species C, reducing the thermodynamic inhibition of species A. We term this a syntrophy interaction.
(TIF)

**S2 Fig. Pollution can render species non-viable.** Our system here consists of three metabolites (the first two of which are supplied) and two species. Species A breaks down metabolite 2 to produce metabolite 3, and species B breaks down metabolite 1 to produce metabolite 3. **A**: When species A is grown on its own it reaches a steady state population. **B**: As species A only breaks down metabolite 2, both metabolite 1 and 3 accumulate. **C**: When species B is added to the system species A is driven to extinction. **D**: In this case, greater accumulation of metabolite 3 occurs as species B breaks down metabolite 1 into it. Species A therefore experiences a greater level of thermodynamic inhibition. As species A and B do not share a substrate this competitive exclusion occurs purely via waste products, we therefore term this a pollution interaction.
(TIF)

**S3 Fig. Interactions high energy case.** Identical plot to Fig 5 but for the high energy supply case ($1.5 \times 10^7$ J mol$^{-1}$).
(TIF)

## Acknowledgments

We thank Tom Clegg, Alexander Christensen, Paul Huxley, Pablo Lechón, and Thomas P. Smith for helpful discussions and comments on the manuscript. We also acknowledge the Physics of Life Network of Excellence for providing a stimulating environment.

## Author Contributions

**Conceptualization:** Jacob Cook, Samraat Pawar, Robert G. Endres.

**Formal analysis:** Jacob Cook.

**Methodology:** Jacob Cook.

**Software:** Jacob Cook.

**Supervision:** Samraat Pawar, Robert G. Endres.

**Visualization:** Jacob Cook.

**Writing – original draft:** Jacob Cook.

**Writing – review & editing:** Jacob Cook, Samraat Pawar, Robert G. Endres.

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
