## [Decision Letter · Decision Letter 0]

28 Jul 2021

Dear Prof. Endres,

Thank you very much for submitting your manuscript "Thermodynamic constraints on the assembly and diversity of microbial ecosystems are different near to and far from equilibrium" for consideration at PLOS Computational Biology.

As with all papers reviewed by the journal, your manuscript was reviewed by members of the editorial board and by several independent reviewers. In light of the reviews (below this email), we would like to invite the resubmission of a significantly-revised version that takes into account the reviewers' comments.

We cannot make any decision about publication until we have seen the revised manuscript and your response to the reviewers' comments. Your revised manuscript is also likely to be sent to reviewers for further evaluation.

Sincerely,

Kiran Raosaheb Patil, Ph.D.

Deputy Editor

PLOS Computational Biology

Kiran Patil

Deputy Editor

PLOS Computational Biology

Reviewer's Responses to Questions

**Comments to the Authors:**

Reviewer #1: The authors construct an artificial model of microbial communities based on random reaction networks for each modeled species governed by thermodynamics, a limited protein pool and either irreversible or reversible reaction kinetics. The inclusion of a protein budget (so far only utilized on an individual species level) and thermodynamics in a community-scale model is novel and has to my knowledge not been attempted yet. They then proceed to investigate community dynamics and interactions through simulations of the model testing different scenarios (high/low energy substrates and reversible/irreversible kinetics). While an approach like this to some extent always begs the question of how representative such an artificial model is of real microbial communities, I find its application very interesting and useful in terms of assessing different modes of how thermodynamics might influence microbial community composition, interactions and dynamics (something that would by very hard/impossible to do based on real microbial communities due to a lack of thermodynamic information and other necessary state variables).

Minor revisions

---------------

* I am sorry if I missed it, but does your model accurately capture that reactions that operate close to equilibrium need very high enzyme levels (thus resulting in high protein expression costs affecting growth rate) to compensate for the low thermodynamic driving force (see [1]). As growth rate only depends on ribosome fraction and energy availability in your model (and not also explicitly on production of "biomass" precursors from substrates at specific rates), I would assume that reactions close to equilibrium would have very low metabolic rates and the cells relying mainly on reactions that are further away from equilibrium to generate energy, while in reality cells would have to maintain certain reaction rates to facilitate biomass precursor production to achieve a certain growth rate? Could you please elaborate on how this might affect the main outcomes of the study?

* I find the manuscript is lacking details on the occurring interactions and the presentation of the results is primarily from a high level perspective. For example, l. 422-424 in the Discussion mentions

... with competition occurring primarily within functional groups, facilitation and syntrophy occurring primarily between functional groups, and pollution occurring both within and between neighbouring functional groups.

but these results are not presented?

* In line with the previous comment, I think it could be worthwhile highlighting concrete examples of how pollution and syntrophy arises in the simulations (maybe in the Supplements)?

* Niebel, Leopold & Heinemann [2] have reported the existence of an upper rate limit on Gibbs energy dissipation (organism specific). I would be interested to hear your take on this result in the light of your simulations. Given the existence of such a limit, would one expect maybe an increased proportion of thermodynamic interaction than the simulations currently result in? Maybe you could add that to or after the paragraph at l. 378-388?

* l. 23, Microbes experience many more physical constraints than just the two constraints mentioned, see [3]

* Are all metabolites consumed/secreted or are there internal-only metabolites also? Based on l. 82, it sounds like all metabolites are accessible to all species at all times? That would be very far from reality? Beyond maintaining concentration gradients, cellular membranes will prohibit many metabolites from freely diffusing into the environment?

Since all metabolites are also external and can thus facilitate thermodynamic inhibition through products? Is that realistic?

* The authors use Michaelis-Menten kinetics synonymously for irreversible kinetics though a reversible Michael-Menten rate equations exists? Maybe clarify that you specifically mean the original rate law developed by Michaelis and Menten?

* eq. 1 and accompanying text, you use \\alpha here to reference metabolites though for the rest of the manuscript you have reserved \\beta for metabolites and \\alpha for reactions.

* l. 198, how connected and functional are those randomly chosen metabolite networks?

* Fig 2b, why do all surviving species reach the same ribosome fraction? Is that expected?

* Reaction efficiency is not clearly defined I think (I am sorry if I overlooked it)

* l. 301-302, low free energy should be recalcitrant and high free energy should be labile, or?

* It is great that you provided all your code on GitHub, but could you maybe highlight in your READMEs main entry points (like files that produces the figures?)

Typos and grammar

-----------------

* Last sentence in abstract, delete "a" in "of a real world"

* MacArthur consumer-resource resource model -> MacArthur consumer-resource model (?)

* While your writing is very good, please check comma usage of introductory clauses to separate them from main clauses, e.g., l. 158, "As \\eta changes, the reaction ...", l. 246, "For all regimes, ...", etc. throughout the manuscript

* l. 189 the the

* l. 343-345, the sentence does not make sense

References

----------

[1] Noor E, Flamholz A, Bar-Even A, Davidi D, Milo R, et al. (2016) The Protein Cost of Metabolic Fluxes: Prediction from Enzymatic Rate Laws and Cost Minimization. PLOS Computational Biology 12(11): e1005167. https://doi.org/10.1371/journal.pcbi.1005167

[2] Niebel, B., Leupold, S. & Heinemann, M. An upper limit on Gibbs energy dissipation governs cellular metabolism. Nat Metab 1, 125–132 (2019). https://doi.org/10.1038/s42255-018-0006-7

[3] Akbari, A., Yurkovich, J.T., Zielinski, D.C. et al. The quantitative metabolome is shaped by abiotic constraints. Nat Commun 12, 3178 (2021). https://doi.org/10.1038/s41467-021-23214-9

Reviewer #2: Please see attachment.

**Have the authors made all data and (if applicable) computational code underlying the findings in their manuscript fully available?**

Reviewer #1: Yes

Reviewer #2: Yes

PLOS authors have the option to publish the peer review history of their article (what does this mean?). If published, this will include your full peer review and any attached files.

Reviewer #1: **Yes: **Nikolaus Sonnenschein

Reviewer #2: No
---

## [Decision Letter · Decision Letter 1]

15 Nov 2021

Dear Prof. Endres,

We are pleased to inform you that your manuscript 'Thermodynamic constraints on the assembly and diversity of microbial ecosystems are different near to and far from equilibrium' has been provisionally accepted for publication in PLOS Computational Biology.

Best regards,

Kiran Raosaheb Patil, Ph.D.

Deputy Editor

PLOS Computational Biology

Reviewer's Responses to Questions

**Comments to the Authors:**

Reviewer #1: The authors have addressed all my concerns adequately.

Reviewer #2: I feel like the authors have done an adequate job addressing my concerns. I recommend publication of the manuscript. My only suggestion is that since PLoS Comp has no word limit that more of the interesting detailed discussion in S9 would benefit from being in the main text.

**Have the authors made all data and (if applicable) computational code underlying the findings in their manuscript fully available?**

Reviewer #1: Yes

Reviewer #2: Yes

PLOS authors have the option to publish the peer review history of their article (what does this mean?). If published, this will include your full peer review and any attached files.

Reviewer #1: **Yes: **Nikolaus Sonnenschein

Reviewer #2: No

---

## [Editor Report · Acceptance letter]

29 Nov 2021

PCOMPBIOL-D-21-01016R1 

Thermodynamic constraints on the assembly and diversity of microbial ecosystems are different near to and far from equilibrium

Dear Dr Endres,

I am pleased to inform you that your manuscript has been formally accepted for publication in PLOS Computational Biology. Your manuscript is now with our production department and you will be notified of the publication date in due course.

With kind regards,

Zsanett Szabo
